# Italian Translation and Validation of the Original ABC Taxonomy for Medication Adherence

**DOI:** 10.3390/healthcare11060846

**Published:** 2023-03-13

**Authors:** Sara Mucherino, Marina Maffoni, Clara Cena, Lucrezia Greta Armando, Marta Guastavigna, Valentina Orlando, Giancarlo Orofino, Sara Traina, Anna Giardini, Enrica Menditto

**Affiliations:** 1CIRFF, Center of Pharmacoeconomics and Drug Utilization, Department of Pharmacy, University of Naples Federico II, 80131 Naples, Italy; sara.mucherino@unina.it (S.M.);; 2Psychology Unit of Montescano Institute, Istituti Clinici Scientifici Maugeri IRCCS, 27040 Montescano, Italy; 3Department of Drug Science and Technology, University of Turin, 10125 Turin, Italy; 4S.C. Malattie Infettive e Tropicali I, ASL Città di Torino, Amedeo di Savoia Hospital, 10149 Turin, Italy; 5Information Technology, Istituti Clinici Scientifici Maugeri IRCCS Pavia, 27100 Pavia, Italy

**Keywords:** Medication adherence, ABC taxonomy, Italian translation, Delphi method, Adherence experts

## Abstract

Medication adherence represents a complex and multifaceted process. Standardized terminology is essential to enable a reproducible process in various languages. The study’s aim was to translate and adapt the original Ascertaining Barriers for Compliance (ABC) Taxonomy on medication adherence, first proposed in 2012, into Italian language. The study was carried out according to the Preferred Methods for Translation of the ABC Taxonomy for Medication Adherence adopted by the ESPACOMP. Key steps included: (1) a systematic literature review using PubMed and Embase according to the PRISMA Guidelines to identify published Italian terms and definitions, and Italian adherence experts; (2) a forward translation of terms and definitions; (3) panelists’ selection; (4) a three-round Delphi survey. From the systematic review, 19 studies allowed detection of 4 terms, 4 definitions and 767 Italian experts. To these, Italian ESPACOMP members and experts though snowball sampling were added. The identified Italian adherence experts received the Delphi questionnaire. The Italian ABC Taxonomy was achieved after three rounds of Delphi survey by reaching at least a moderate consensus on unambiguous naming and definition of medication adherence-related terms. The Taxonomy is intended to be used in research, academic, and professional fields in order to harmonize adherence terminology and avoid confusion in comparing research findings.

## 1. Introduction

In recent years, there has been a rapid increase in scientific interest in patient medication adherence. The literature is growing, describing the pervasiveness of poor medication adherence, which experts recognize as a significant public health concern mainly related to adverse health care outcomes and increased health care costs [1,2]. For instance, experts estimate that poor adherence is causing €125 billion in avoidable hospitalizations, emergency care, and outpatient visits in Europe and $105 billion in the United States per year, and this expenditure is going to increase in the next few years [3,4,5,6,7,8,9]. Many subjective, relational, and environmental aspects may contribute to non-adherence. On one hand, cognitive impairment, previous negative experiences with medications, poor health literacy, beliefs, and fears of side effects, and drug–drug interactions may threaten medication adherence. On the other hand, a lack of social and family support and a poor alliance between the clinician and the patient may also undermine medication adherence [10,11,12,13,14,15]. Moreover, complex drug characteristics (such as tablet size/dosage unit size, time and method of drug intake, pill burden) and difficulties in accessing healthcare services may also hinder medication adherence [16,17,18].

Thus, medication adherence represents a complex and multifaceted process, and understanding and improving it are an urgent imperative in the present and future health care landscape, considering the increase in multimorbidity and population aging [19]. Standardized terminology is essential to fully understand the medication adherence phenomenon and to enable a reproducible process in various languages, aiming to compare the results obtained from medication adherence studies conducted worldwide [20]. In this scenario, the ABC (Ascertaining Barriers to Compliance) project was created as a European initiative consisting of research groups operating in the field of adherence to medications funded by the European Commission, Seventh Framework Programme. To respond to this need, the ABC Taxonomy was first proposed by ESPACOMP, the International Society for Medication Adherence (https://www.espacomp.eu/project/abc-taxonomy/ (accessed on 7 May 2022)), in 2012 with the aim of promoting consistency and quantification of the terms used to describe [21].

Briefly, this conceptualization describes adherence as a multifaceted process developing through phases over time, which may totally or partially fail because of late initiation or non-initiation (initiation), suboptimal pursuance and perdurance (implementation and persistence, respectively), or early interruption (discontinuation) of a certain drug treatment. Thanks to the growing interest in the ABC Taxonomy in scientific research and to its implications for improving medication adherence in daily practice, the ABC Taxonomy may be considered a promising and useful model to conceptualize and study medication adherence [22,23]. The Taxonomy was first published in English and subsequently translated into French and German with the aim of harmonizing terminology across languages and further increasing comparability in scientific research [24]. Thus, it is necessary to increase the number of languages in which to standardize and validate the terminology related to medication adherence, with the ultimate goal of eradicating ambiguity in adherence research. To do so, a shared document was published by ESPACOMP describing methods to be adopted for the translation of the ABC adherence taxonomy, into other languages, namely Preferred Methods for Translation of the ABC Taxonomy for Medication Adherence [25]. These methods includes several harmonized key steps, such as a literature search, forward translation of terms/definitions, panelists’ selection, and Delphi survey to reach consensus in the target language [25]. Actually, in the Italian setting there is still an unmet need for a unified taxonomy on medication adherence research measures and terminology. This addresses the lack of consistency and clarity in medication adherence national research, which can lead to confusion and difficulty in comparing and synthesizing findings across studies. In this vein, the present study aimed to translate and adapt the original ABC Taxonomy on medication adherence into the Italian language through translation of the related terms and definitions.

## 2. Materials and Methods

The present study was carried out according to the Preferred Methods for Translation of the ABC Taxonomy for Medication Adherence adopted by the ESPACOMP [25] for the translation of the ABC Taxonomy, originally described in English by Vrijens et al. [21], into other languages. The Delphi method was chosen as the preferred methodology to achieve consensus on the terminology [26].

The key steps included: (1) bibliographic research to identify key papers on medication adherence in the Italian language in order to identify published Taxonomy terms and definitions in Italian, and to identify Italian adherence experts; (2) a preliminary translation of the terms and their definitions; (3) the selection of the panelists; (4) a Delphi survey (design and administration). All the steps described above were divided into an operational phase, carried out by 4 researchers, and the supervisory phase carried out by 5 other researchers. The entire process is graphically shown in Figure 1.

### 2.1. Literature Search

A systematic review was carried out according to the PRISMA 2020 (Preferred Reporting Items for Systematic reviews and Meta-Analyses) statement guidelines to identify Italian studies published on medication adherence [27], to describe how the ABC Taxonomy terms and definitions were defined in Italian, and to identify Italian experts in the area (Appendix A). The review was prospectively registered in PROSPERO—the International Prospective Register of Systematic Reviews (registration code: CRD42020212909).

Two scientific databases, PubMed (via Medline) and EMBASE (via Ovid), were queried from 2012—the ABC Taxonomy publication year—to July 2020. The reason for the selection of these databases was related to their: (i) Comprehensive coverage in the field of biomedical research indexing thousands of journals in the field of medicine, nursing, pharmacy, and other health-related disciplines, and were likely to contain a substantial portion of the relevant literature on medication adherence in the Italian language; (ii) Language specificity allowed for language-specific searches, which was particularly relevant in this case as the focus was on Italian language research papers; (iii) Established quality standards with rigorous quality standards for the inclusion of articles; (iv) Common practice in the field of medicine and health sciences. This enabled the systematic review study to follow a standardized and widely accepted methodology, which enhances the credibility and replicability of the study. Inclusion criteria were based on the identification of original articles published in peer-reviewed journals and available in the Italian language. An Italian language filter was set. Moreover, the exclusion criteria consisted of books, theses, research protocols, conference proceedings, abstracts, posters, and research studies not available in the Italian language.

The search strategy combined principally the 7 terms mainly related to medication adherence research (Medication adherence; Initiation; Implementation; Discontinuation; Persistence; Adherence management; Adherence-related science) [21], as reported from the original ABC Taxonomy on Medication Adherence, with all their pertaining synonyms (Adherence; Compliance; Patient compliance; Treatment adherence; Medication compliance; Medication persistence; Treatment compliance; Adhesion; Interruption) [28]. These terms were searched as MeSH Term or Emtree. The Boolean operators AND/OR were used to combine searches and obtain the two final syntaxes. Entire search strategy is available in Appendix A. The screening process was organized in two phases: title and abstract screening and full-text screening. Four researchers (S.M, M.M., S.T., L.G.A.) independently screened titles and abstracts and selected them for the next step. The same four researchers independently screened full-texts included in the analysis for their eligibility according to shared inclusion/exclusion criteria. The other authors participated in the screening process and resolved any disagreements regarding some records to reach a consensus. After making a shared decision, they identified the final number of full-text records to include. Then, they extracted the following information from each study: title, authors’ names, corresponding author’s name, corresponding author’s email, year of publication, journal, and the ABC Taxonomy term and definition used in each study.

### 2.2. Forward Translation

A single forward translation from English to Italian of terms and definitions which were not found into the systematic review process was completed by 4 native Italian researchers in the field (S.M, M.M., S.T., L.G.A.) who were also fluent in English. Terms and definitions translated were discussed and confirmed by other native Italian researchers fluent in English (E.M., A.G., C.C., M.G., V.O., G.O.). Country-specific adaptions were performed where needed in view of facilitating the implementation of the terms and definitions into Italian practice [24]. No backward translation was carried out because the experts involved in the three-round Delphi survey in the next phase made implicit backward/forward translations in expressing their views and opinions.

### 2.3. Selection of the Panelists

In this stage, we identified Italian adherence experts to participate in the survey using the Delphi method. We enrolled panelists who were Italian natives fluent in English and who had interests in the fields of medication adherence research and education. These panelists were selected as follows:Italian ESPACOMP (International Society for Medication Adherence) members;Corresponding authors of Italian articles selected by systematically reviewing papers identified through PubMed and Embase;“Snowball sampling”: a non-probability sampling technique in which enrolled study subjects recruit other subjects among their local network (personal contacts).

Panelists’ occupations were categorized in their professional field, as follows: biology, biostatistics, economics/health management, nursing, medicine (GPs and specialists), psychology, pharmaceutical sciences (community, clinical, hospital pharmacists, academia, etc.), social sciences (rehabilitation, researchers, etc.), and an open field including all other professions not included in the previous ones (patient representative, scientific information, clinical risk, laboratory technicians, etc.). The invitation to participate in the study was sent by e-mail. The consent to participate was properly requested according to the EU General Data Protection Regulation 2016/679 (GDPR).

### 2.4. Delphi Survey

We sent a three-round Delphi survey by email to the identified experts, and we aggregated their responses and shared them with the group after each round. The e-survey consisted of two parts: the first part contained general information, such as consent to data processing, reference email for sending subsequent rounds, and professional field. The second part contained various proposals for the Italian translation of the 7 ABC Taxonomy terms and definitions resulting from the systematic review and/or suitably integrated when missing. The three-round Delphi survey is shown in Appendix A.

The objective of the Delphi survey was to achieve an unambiguous response through consensus. In line with the previous literature, consensus on the translated items was defined according to the following acceptance rates:

Moderate consensus (50–75% acceptance rate): This level of consensus was achieved when a majority of the participants expressed their agreement on a specific Italian translation of an ABC Taxonomy term/definition. Specifically, at least 50% of the participants were in agreement on the translation; Disagreement (<50% acceptance rate): An acceptance rate of less than 50% was considered to be a low level of consensus, indicating that there was disagreement among the participants regarding the Italian translation of an ABC Taxonomy term/definition. This meant that less than half of the experts consulted in the Delphi survey agreed with the translation; Consensus (>75–90% acceptance rate): This level of consensus was achieved when a substantial majority of the participants expressed agreement on a specific Italian translation of an ABC Taxonomy term/definition. Hence, at least 75% of the participants were in agreement on the translation; Strong consensus (>90% acceptance rate): This level of consensus was achieved when an overwhelming majority of the participants expressed agreement on a specific Italian translation of an ABC Taxonomy term/definition, thus, more than 90% of the participants were in agreement on the translation [24].

Panelists’ responses were iterative in batches, thus eliminating influence. The Delphi survey was carried out by e-mail in three different rounds containing the active link to the survey without a password request. Google forms was used to create the online survey rounds. Three reminders were sent at the frequency of 2–3 weeks for each round. The survey was preceded by a pilot interview among 6 junior researchers in order to re-examine the questions and to check their clarity.

Round-1: The items in Round-1 derived from Italian terms and definitions of the ABC Taxonomy resulted from the studies included in the literature review process described above. Questions were sent to the panel of experts with the published definitions (if available); definitions absent in the publications were derived from a native Italian speaker translation and a free text field. Panel members were asked to select 1 preferred item (single choice) or to propose new terms and definitions in a free text field. Items with an acceptance rate <10% were discarded from the next round.

Round-2: A second set of items based on previous answers was sent to the panelists who responded to Round-1. Terms and definitions obtained from Round-1 and the level of agreement were indicated. Definitions were grouped together and similar formulations were reduced to one comprehensive statement. New terms and definitions were allowed to be proposed in a free text field. Items with an acceptance rate <10% and >75% were not integrated into the next round.

Round-3: The last set of questions based on previous answers was sent to the panelists who responded to Round-2. Terms and definitions obtained from Round-2 and their relative level of consensus were proposed.

## 3. Results

### 3.1. Systematic Literature Process

During the systematic review process, we identified 79 Italian papers on medication adherence through database searching. After removing duplicates, we selected 72 articles. We included a total of 19 studies that met the inclusion criteria in the analysis [29,30,31,32,33,34,35,36,37,38,39,40,41,42,43,44,45,46] (Appendix A). Of the included studies, 18 (94.7%) mentioned “adherence to treatment”; 10 (52.6%) mentioned the term “discontinuation”; 7 (36.8%) cited the term “persistence”; 2 (15.8%) named the term “initiation”. We did not detect any other ABC Taxonomy adherence-related terms (Appendix A). Regarding the terms’ definitions, five studies (26.3%) included a definition of the term “adherence to treatment”; three studies (15.8%) defined the term “implementation”; ten studies (21.1%) defined the term “discontinuation”; and four studies (21.1%) defined the term “persistence”. In more detail, we detected several different Italian translations for citing and defining each term from the 19 included studies, which are reported in Appendix A. We used the adherence-related Italian studies from the systematic review to identify the Italian translations of terms and definitions to include in Round-1 of the Delphi survey. These detected options were: 9 for the “Adherence to medication” term and 5 for its definition; 4 for the term “Initiation”; 3 for the definition of “Implementation”; 5 for the term “Discontinuation” and 4 for its definition; 3 for the term “Persistence” and 4 for its definition. As for the terms “Implementation”, “Management of adherence”, and “Adherence-related science”, no translations were detected from the studies analyzed, so forward translation was performed by the researchers.

### 3.2. Delphi Survey

Overall, 767 Italian adherence experts received the Round-1 online questionnaire. Round-1 reached a response rate of 22%, (number of panelists: 165; number of proposed items: 30); Round-2 reached a response rate of 67% (number of panelists: 110; number of proposed items: 29); Round-3 reached a response rate of 80% (number of panelists: 88; number of proposed items: 27) (Figure 2).

The most common professional fields of panelists were pharmaceutical sciences (Round-1: 36%; Round-2: 38%; Round-3: 38%) and medicine (Round-1: 32%; Round-2: 33%; Round-3: 34%) (Figure 3).

The Italian-speaking panelists reached a moderate consensus for all the terms and definitions reaching at least 50–75% of agreement. For the term and definition of “Management of adherence”, a higher consensus was reached (75–90% of agreement). Table 1 and Table 2 show the consensus rate reached for the seven terms and definitions of the translations at each Delphi round, respectively.

In Round-2, a moderate consensus was reached for the terms “Inizio della terapia farmacologica” (58%), “Persistenza alla terapia farmacologica” (61%), “Interruzione della terapia farmacologica” (61%), and a consensus was reached for the term “Gestione dell’aderenza terapeutica” (81%). In Round-3, a moderate consensus was reached for the terms “Aderenza alla terapia farmacologica” (61%), “Effettiva assunzione della terapia al dosaggio prescritto” (64%), and “Scienza rivolta allo studio dell’aderenza” (65%) (Figure 4).

Proposed definitions of “Inizio della terapia farmacologica”, “Interruzione della terapia farmacologica”, “Gestione dell’aderenza terapeutica”, and “Scienza rivolta allo studio dell’aderenza” reached acceptance rates between 51 and 62% (moderate consensus) in Round-1 and continued to be selected in subsequent rounds despite new proposals. In Round-2, we achieved a final moderate consensus for “Effettiva assunzione della terapia al dosaggio prescritto” (61%), “Interruzione della terapia farmacologica” (66%), and “Scienza rivolta allo studio dell’aderenza” (57%). In the last round (Round-3), definitions of the terms “Aderenza alla terapia farmacologica”, “Inizio della terapia farmacologica”, and “Persistenza alla terapia farmacologica” reached a moderate consensus (64%), while “Gestione dell’aderenza terapeutica” definition translation reached a consensus (75%) (Figure 5). This analysis produced an Italian version of the ABC Taxonomy that includes the following seven Italian terms: (1) “Aderenza alla terapia farmacologica” (Round-3, 61%), defined as “Il processo attraverso cui i pazienti assumono i loro farmaci come prescritto” (Round-3, 64%); (2) “Inizio della terapia farmacologica” (Round-2, 58%), defined as “Il processo inizia con l’inizio del trattamento, quando il paziente assume la prima dose di un farmaco prescritto” (Round-3, 64%); (3) “Effettiva assunzione della terapia al dosaggio prescritto” (Round-3, 64%), defined as “Il processo continua con il raggiungimento del regime di dosaggio farmacologico prescritto, definito come la misura in cui il dosaggio effettivamente assunto dal paziente corrisponde a quello prescrittogli, dall’inizio della terapia fino all’assunzione dell’ultima dose” (Round-2, 61%); (4) “Persistenza alla terapia farmacologica” (Round-2, 61%), defined as “La persistenza è il periodo di tempo tra l’inizio della terapia e l’ultima dose assunta immediatamente precedente l’interruzione” (Round-3, 64%); (5) “Interruzione della terapia farmacologica” (Round-2, 61%), defined as “L’interruzione definisce la fine della terapia, quando la dose successiva da assumere viene omessa e non vengono più assunte altre dosi” (Round-2, 66%); (6) “Gestione dell’aderenza terapeutica” (Round-2, 81%), defined as “È il processo di monitoraggio e sostegno dell’aderenza alla terapia dei pazienti da parte dei sistemi e degli operatori sanitari, dei pazienti e delle loro reti sociali. L’obiettivo della gestione dell’aderenza è quello di ottenere, da parte dei pazienti, il miglior utilizzo possibile dei farmaci adeguatamente prescritti, al fine di rendere massimo il beneficio e minimo il rischio di danno” (Round-2, 75%); (7) “Scienza rivolta allo studio dell’aderenza” (Round-3, 65%), defined as “Questo elemento include le discipline che mirano a comprendere le cause o le conseguenze della differenza tra l’esposizione prescritta ai farmaci (cioè prevista dal medico prescrittore) e l’esposizione effettiva. La complessità di questo campo di ricerca, così come la sua ricchezza, derivano dal fatto che esso opera oltre i confini di diverse discipline, tra le quali, ma non solo, la medicina, la farmacia, le scienze infermieristiche, le scienze comportamentali, la sociologia, la farmacometria, la biostatistica e l’economia sanitaria” (Round-2, 57%). Table 3 shows the complete Italian translation of all ABC Taxonomy terms and definitions, as compared to the original English Taxonomy that was achieved.

## 4. Discussion

We used a systematic review process of the Italian literature and a subsequent Delphi survey to define the Italian ABC Taxonomy and reach a consensus on the unambiguous naming and definition of terms related to the medication adherence process. To the best of our knowledge, this is the first study reporting findings for advancing the harmonization of Italian medication adherence definition by promoting clear and shared terminology to standardize research in the field. This issue is crucial as most of the terms still in use today regarding medication adherence do not have a clear or direct translation in the different European languages [47,48], which can lead to misunderstandings and hinder comparability between studies and implementation in clinical daily practice [49,50].

These considerations support the need to validate the ABC Taxonomy at a local level, as already performed in German and French [24]. In the Italian setting, different terminologies in various fields of action have so far rendered communication difficult, both in research and in the implementation of practical actions. The results of the Delphi among Italian experts confirmed this discrepancy, requiring three rounds to reach a consensus for all terms and definitions related to medication adherence. Specifically, findings indicated that six/seven terms, such as “Adherence to medication”, “Initiation”, “Implementation”, “Persistence”, “Discontinuation”, and “Adherence-related science”, reached at most a moderate consensus, i.e., at least 50–75% of the experts agreed with the same translation/definition.

One of the most sensitive challenges has been to find an Italian term to effectively translate “Adherence”, which differs from “Persistence”. This point can be due to the fact that, for more than two decades, the term adherence has been confused in Italian language with the terms “Compliance”, “Adhesion”, and “Persistence” (translated as “Compliance”, “Adesione”, “Persistenza”, respectively) [28,51]. Therefore, this enabled a greater number of synonyms for a single word to be identified both in the systematic review process and in the questioning of respondents. The term “adherence” was preferred over “compliance” in Italy, as it encompasses the patient’s involvement in the treatment process and willingness to follow the healthcare provider’s advice. While “Persistence” was used to describe the duration of medication use, particularly for chronic conditions, “Concordance” was used to describe a collaborative approach to medication management that emphasizes communication, mutual respect, and shared decision-making to improve adherence and treatment outcomes. This different terminology attitude may help explain why a lower consensus rate (moderate) was identified for most of the terms [24]. In addition to these considerations on the specific characteristics of the Italian versus English language, most of the terms reached a moderate consensus despite the German and French translations, where higher levels of agreement were reached [24]. This could explain how the linguistic contexts may consider the same concepts differently. Hence, it is noteworthy that, following the three Delphi rounds performed, the Italian adherence experts reached fairly high levels of consensus on the choice of the terms “Aderenza alla terapia farmacologica” (61%), “Inizio della terapia farmacologica” (58%), “Persistenza alla terapia farmacologica” (61%), and “Interruzione della terapia farmacologica” (61%). This underlines the fact that the experts agree that the terms relating to the definitions of adherence process phases, e.g., Medication Adherence, Initiation, Persistence, and Discontinuation, are purely drug therapy-related events. In even more detail, the experts reached a consensus of 64% in translating the term “Implementation” with “Effettiva assunzione della terapia al dosaggio prescritto”; in this case, the experts considered it appropriate to specify that implementation is linked to the prescribed dosage, since this process describe the dosing history, so the extent to which a patient’s actual dosing corresponds to the prescribed dosing regimen, from initiation until the last dose is taken. Despite this, a strong consensus (81%) was reached at Delphi Round-2 for the translation of the term “Adherence management” into “Gestione dell’aderenza terapeutica”. For this term alone, “drug therapy” is not specified, but adherence management is understood more broadly, as the management of an entire therapy-related process. Therefore, these findings address that harmonization in this field is an urgent imperative as it will allow adherence researchers to communicate effectively and unambiguously.

To sum up, this study could suggest the promotion of a unique adherence Taxonomy which could be applied in real life clinical practice contexts.

Overall, providing the clinical and scientific community with a shared terminology on adherence is particularly crucial in the actual and future health care landscape. Indeed, it is widely recognized in the actual literature that success in medication adherence-behavior requires a coordinated intervention by the main actors involved (i.e., patient, general practitioner and specialist, pharmacist, paramedic, psychologist/psychotherapist, family member, health authorities, pharmaceutical industry) [52,53,54], combined with extensive awareness-raising initiatives and dissemination of the basic principles underlying strategies to assess and monitor over time the non-adherence to treatments. To reach this aim, an unambiguous and univocal communication is necessary [55]. Ineffective communication between health care professionals and chronically ill patients could further compromise the patients’ understanding of their disease, also influencing their adherence behavior leading to potential complications [56].

Active patient engagement in all aspects related to the management of their health is crucial for fostering better disease knowledge and effective communication with healthcare professionals. While the accurate terminology on medication adherence disseminated by the review and the Italian translation of the ABC Taxonomy may have a positive impact on chronic patients’ self-efficacy and empowerment, future steps must involve effectively involving patients in the Taxonomy decision-making process. The increasing utilization of patient-reported outcomes (PRO) and health-related quality of life metrics (HR-QOL) in clinical practice and chronic conditions’ management needs a clear and unique vocabulary in questionnaires and/or surveys. Any intervention directed to the improvement of patient health literacy and the capability to communicate about health conditions, disease symptoms and progression, and drug prescriptions, could help the achievement of trust in clinicians and their prescribed therapies. In this way, more “expert” patients could gain self-efficacy, which represents an essential skill to effectively manage their condition, organizing and implementing a set of actions needed to cope effectively with complex therapeutic regimens and, through the activation of cognitive, emotional, relational, and behavioral resources, gain empowerment, acquiring an active and mature role in controlling future events and expectations [57].

As the research on medication adherence has evolved over time, it has encompassed various areas including biomedical, technological, sociological, and behavioral perspectives, each with its own distinct concepts [21]. Given that the ABC Taxonomy is a widely recognized model that views adherence as a process with specific phases, it would be worthwhile to extend this terminology beyond the medical and pharmaceutical fields to the behavioral realm. Reaching one shared and common terminology to foster adherence can also play a pivotal role in case of different types of prescriptions. In this regard, the literature has already unveiled improved medical and functional outcomes when the patient shows a satisfactory adherence to non-pharmacological treatments, such as interventions focusing on rehabilitation (e.g., physical and/or cognitive) or promotion of a positive lifestyle (e.g., no smoking, limited use of alcohol) [58,59,60,61,62]. Thus, as a future recommendation, there is the suggestion to explore the use of a standardized and shared terminology of all facets of adherence, and also in the case of behavioral treatment. Hence, a shared and standardized adherence terminology can influence how medication adherence behavior is understood, measured, and addressed in clinical practice. Choosing the right terminology and understanding the nuances of each term can help healthcare providers and researchers more accurately assess medication adherence and develop effective strategies to improve it. In this regard, ABC Taxonomy could be a promising model that should be validated and further explored to support and foster adherence to psychological/psychotherapeutic prescriptions, as well as to other behavioral recommendations (e.g., lifestyle, rehabilitation interventions).

## 5. Strengths and Limits

The present study has several strengths to pinpoint. We adopted a systematic approach to identify eligible experts to include in the survey. Thereafter, the conduct of the systematic review in accordance with the PRISMA Statement also allowed the unveiling of all Italian adherence-related terms present in the literature so far, which were included in the Delphi questionnaire. Thus, a validated methodology was used to perform the survey for reaching consensus, the Delphi approach, already tested elsewhere in order to guarantee the validity and comparability of results [63].

However, certain limitations must be recognized. First, the main point of debate is related to the nature of the Delphi technique, in particular concerning its reliability and validity [25]. An example is that terms and definitions rated with <10% acceptance were excluded and we cannot theoretically exclude that these discarded voices might have won the consensus process in a later round. Moreover, if the response rate had been higher and more varied, it is not certain that we would have received the same results. This issue was already discussed and could be overcome by considering Lincoln and Guba’s criteria for qualitative studies which are credibility (truthfulness) [64], fittingness (applicability), auditability (consistency), and confirmability. Regarding the validity concern, the involvement of participants who have recognized expertise in the same topic may help with increasing the Delphi content’s validity [65], and the use of consecutive rounds can help to increase the concurrent validity. Nonetheless, it has to be stated that the results’ validity will be ultimately affected by the response rates.

## 6. Conclusions

This study provides the Italian-translated ABC Taxonomy on Medication Adherence obtained through a multi-step standardized process involving Italian experts. The Italian Taxonomy is intended to be used in the research, academic, and professional fields in order to harmonize adherence terminology and avoid confusion in comparing research findings. As a future overview, validation of the Italian-translated ABC Taxonomy on Medication Adherence could be useful to ensure that it is a reliable and valid tool for use in Italian-speaking populations. This could involve testing the tool in different settings and with different populations to ensure that it produces consistent and meaningful results. Finally, these findings could represent the key point to explore the use of a standardized and shared terminology of all facets of adherence, extending to behavioral contexts too, as well as to a specific Taxonomy for use in real clinical practice.

## Figures and Tables

**Figure 1 healthcare-11-00846-f001:**
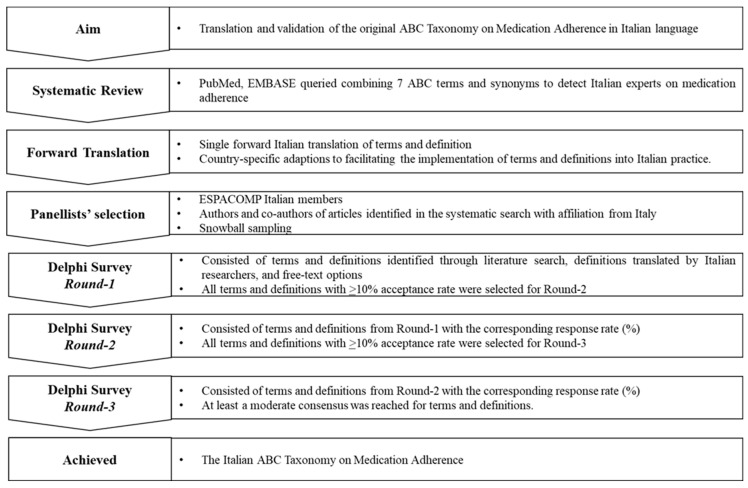
Key steps of the study process.

**Figure 2 healthcare-11-00846-f002:**
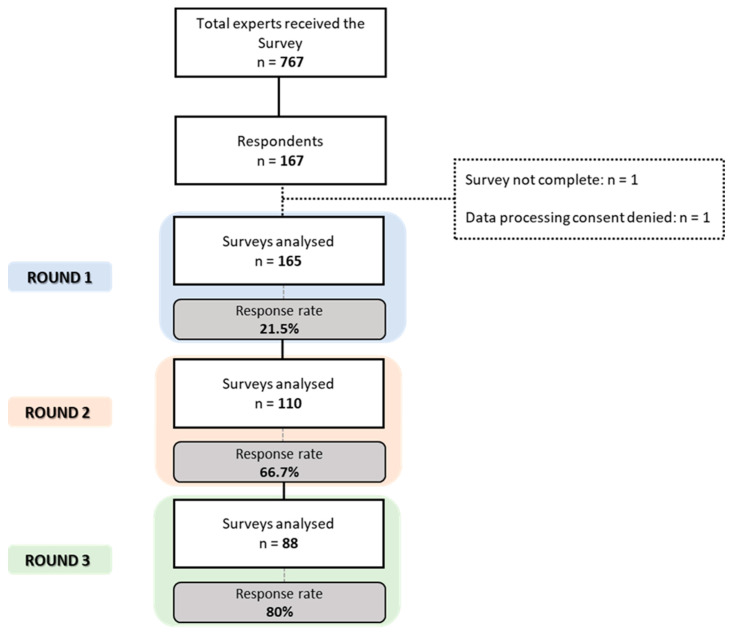
Identification process of Italian experts in Adherence research field.

**Figure 3 healthcare-11-00846-f003:**
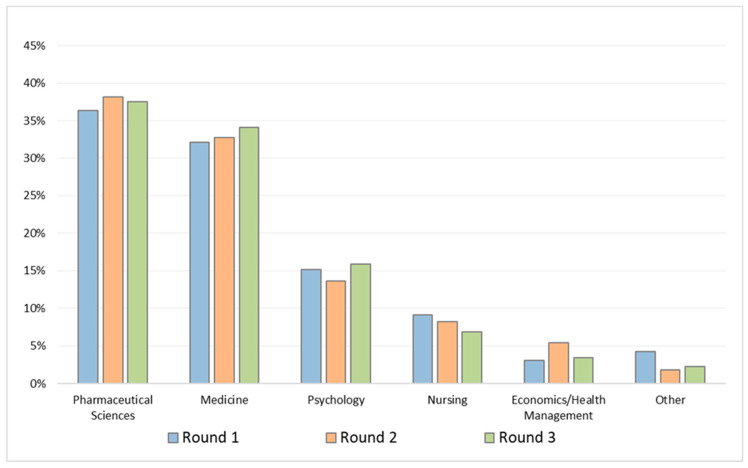
Professional field of the Italian-speaking experts in Rounds 1, 2, and 3.

**Figure 4 healthcare-11-00846-f004:**
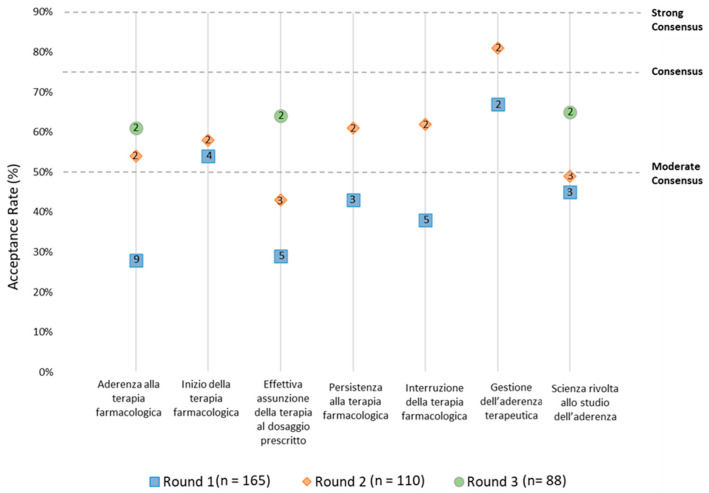
Preferred Italian terms per Delphi round with the number of proposed items in the icon (numbers in the boxes are representing the amount of different translations proposed for each round).

**Figure 5 healthcare-11-00846-f005:**
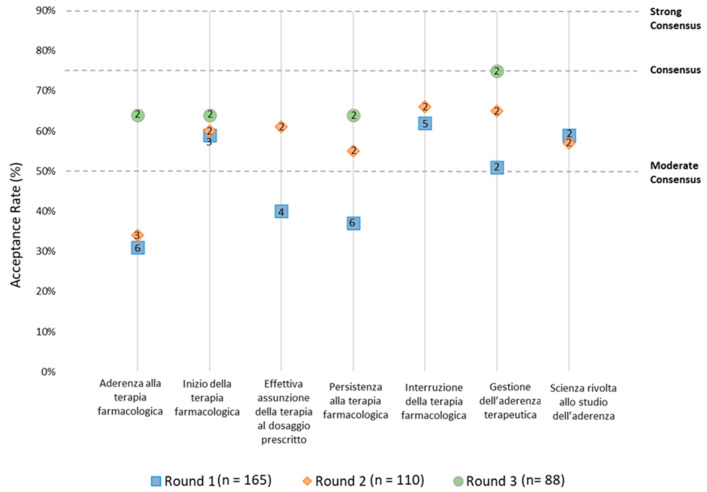
Preferred Italian definitions per Delphi round with the number of proposed items in the icon (numbers in the boxes are representing the amount of different translations proposed for each round).

**Table 1 healthcare-11-00846-t001:** Round needed for consensus reaching for Italian translation of terms.

English Taxonomy	ROUND 1		ROUND 2		ROUND 3	
	Italian Term		Italian Term		Italian Term	
	Options Number	Consensus Obtained	Options Number	Consensus Obtained	Options Number	Consensus Obtained
Adherence to medication	9	No (28%)	2	Moderate consensus (54%)	2	**Moderate** **consensus** **(61%)**
Initiation	4	Moderate consensus (54%)	2	**Moderate** **consensus** **(58%)**	-	-
Implementation	5	No (29%)	3	No (43%)	2	**Moderate** **consensus** **(64%)**
Persistence	3	No (43%)	2	**Moderate** **consensus** **(61%)**	-	-
Discontinuation	5	No (38%)	2	**Moderate (62%)**	-	-
Management of adherence	2	Moderate consensus (67%)	2	**Consensus (81%)**	-	-
Adherence-related sciences	3	No (45%)	3	No (49%)	2	**Moderate** **consensus** **(65%)**

Notes: Final round is the one in which the definition has reached the highest level of consensus (in bold).

**Table 2 healthcare-11-00846-t002:** Round needed for consensus reaching for Italian translation of definitions.

English Taxonomy	ROUND 1		ROUND 2		ROUND 3	
	Italian Definition		Italian Definition		Italian Definition	
	Options Number	Consensus Obtained	Options Number	Consensus Obtained	Options Number	Consensus Obtained
Adherence to medication	6	No (31%)	3	No (34%)	2	**Moderate** **consensus** **(64%)**
Initiation	3	Moderate consensus (59%)	2	Moderate consensus (60%)	2	**Moderate** **consensus** **(64%)**
Implementation	4	No (40%)	2	**Moderate** **consensus** **(61%)**	-	-
Persistence	6	No (37%)	2	Moderate consensus (55%)	2	**Moderate** **consensus** **(64%)**
Discontinuation	5	Moderate consensus (62%)	2	**Moderate** **consensus** **(66%)**	-	-
Management of adherence	2	Moderate consensus (51%)	2	Moderate consensus (65%)	2	**Consensus (75%)**
Adherence-related sciences	2	Moderate consensus (56%)	2	**Moderate** **consensus** **(57%)**	-	-

Notes: Final round is the one in which the definition has reached the highest level of consensus (in bold).

**Table 3 healthcare-11-00846-t003:** Italian translation of the ABC taxonomy and corresponding definitions including the acceptance rate (%) after the corresponding Delphi round (2nd, 3rd).

English Taxonomy	English Definition	Italian Taxonomy	Italian Definition
Adherence to medication	The process by which patients take their medications as prescribed	Aderenza alla terapia farmacologica (3rd, 61%)	Il processo attraverso cui i pazienti assumono i loro farmaci come prescritto (3rd, 64%)
Initiation	The process starts with initiation of the treatment, when the patient takes the first dose of a prescribed medication	Inizio della terapia farmacologica (2nd, 58%)	Il processo inizia con l’inizio del trattamento, quando il paziente assume la prima dose di un farmaco prescritto (3rd, 64%)
Implementation	The process continues with the implementation of the dosing regimen, defined as the extent to which a patient’s actual dosing corresponds to the prescribed dosing regimen, from initiation until the last dose is taken	Effettiva assunzione della terapia al dosaggio prescritto (3rd, 64%)	Il processo continua con il raggiungimento del regime di dosaggio farmacologico prescritto, definito come la misura in cui il dosaggio effettivamente assunto dal paziente corrisponde a quello prescrittogli, dall’inizio della terapia fino all’assunzione dell’ultima dose (2nd, 61%)
Persistence	Discontinuation marks the end of therapy, when the next dose to be taken is omitted and no more doses are taken thereafter	Persistenza alla terapia farmacologica (2nd, 61%)	La persistenza è il periodo di tempo tra l’inizio della terapia e l’ultima dose assunta immediatamente precedente l’interruzione (3rd, 64%)
Discontinuation	Persistence is the length of time between initiation and the last dose, which immediately precedes discontinuation	Interruzione della terapia farmacologica (2nd, 61%)	L’interruzione definisce la fine della terapia, quando la dose successiva da assumere viene omessa e non vengono più assunte altre dosi (2nd, 66%)
Management of adherence	It is the process of monitoring and supporting patients’ adherence to medications by health care systems, providers, patient, and their social networks. The objective of management of adherence is to achieve the best use by patients, of appropriately prescribed medicines, in order to maximize the potential for benefit and minimize the risk of harm	Gestione dell’aderenza terapeutica (2rd, 81%)	È il processo di monitoraggio e sostegno dell’aderenza alla terapia dei pazienti da parte dei sistemi e degli operatori sanitari, dei pazienti e delle loro reti sociali. L’obiettivo della gestione dell’aderenza è quello di ottenere, da parte dei pazienti, il miglior utilizzo possibile dei farmaci adeguatamente prescritti, al fine di rendere massimo il beneficio e minimo il rischio di danno (3rd, 75%)
Adherence-related sciences	This element includes the disciplines that seek understanding of the causes or consequences of differences between the prescribed (i.e., intended) and actual exposures to medicines. The complexity of this field, as well as its richness, results from the fact that it operates across the boundaries between many disciplines, including but not limited to medicine, pharmacy, nursing, behavioral science, sociology, pharmacometrics, biostatistics, and health economics	Scienza rivolta allo studio dell’aderenza (3rd, 65%)	Questo elemento include le discipline che mirano a comprendere le cause o le conseguenze della differenza tra l’esposizione prescritta ai farmaci (cioè prevista dal medico prescrittore) e l’esposizione effettiva. La complessità di questo campo di ricerca, così come la sua ricchezza, derivano dal fatto che esso opera oltre i confini di diverse discipline, tra le quali, ma non solo, la medicina, la farmacia, le scienze infermieristiche, le scienze comportamentali, la sociologia, la farmacometria, la biostatistica, e l’economia sanitaria (2nd, 57%)

## Data Availability

Access to the data is allowed. Requests to access the datasets should be directed to the corresponding author, email: enrica.menditto@unina.it.

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
