# Peer review of "Italian Translation and Validation of the Original ABC Taxonomy for Medication Adherence"

_healthcare, 2023, doi:10.3390/healthcare11060846_

Round 1

Reviewer 1 Report

The manuscript entitled " Italian translation and validation of the original ABC Taxonomy for Medication Adherence." translated the ABC Taxonomy on Adherence into Italian through a multi-step standardized process, promoting the normalization of the use of ABC Taxonomy in the research, academic and professional fields in Italy. However, some problems must be solved before this manuscript is considered for publication.

1. In part 1, INTRODUCTION, the authors are suggested to briefly introduce the methods used and the results obtained in this study.

2. Why was the study only included in two scientific databases, PubMed and EMBASE, and not the Web of Science?

3. In part 2.4, DELPHI SURVEY, the definition of the consensus on the translated items of >75-90% acceptance rate was lacking. In addition, it would be better to understand Table 1 and Table 2 if the authors could provide the definition of the consensus on the translated items of <50% acceptance rate.

4. In Table 1 and Table 2, Why some acceptance rates that were 50–75% was defined as “No” rather than “Moderate”?

5. In CONCLUSION part, the authors are suggested to provide some suggestions for the relevant future research.

Reviewer 2 Report

Thank you for allowing me to review this manuscript. This study describes the translation of the ABC adherence taxonomy to the Italian language. Italian is a commonly used language worldwide and represents an important manuscript. The methodology was very well done. There could be a few improvements to facilitate understanding. 

Throughout: There is a lot of passive voice used throughout the manuscript (e.g., 19 studies allowed detection of 4 terms) instead of active voice (e.g., 19 studies identified 4 terms, or something similar). Recommend using more active voice throughout. 

Introduction:

-lines 44-47: "On one hand, medication adherence may be threatened...", nothing is cited here. Please include some citations for this statement.

-lines 71-78: It would be helpful for the authors to place this study in more context for the Italian setting. For instance, how many italian speakers are there? Basically, besides doing this study just to increase the number of languages, why is this study significant (similar to what was done in the discussion)? One to two sentences would suffice.

Methods

-The results mention the type of experts that were included in the delphi technique. It would be helpful to know how the authors categorized pharmaceutical scientists and what types of experts that involves. Is medicine exclusively physicians or did it include other types of health care professionals, such as physician assistants/associates? Who consisted of the 'Other' type? Were psychologists physicians or counselors? What was the level of training for all involved, including nurses?

Reviewer 3 Report

line 40: reviewer would add references

line 104: explain why these two databases were chosen and not others

line 105: it would be good to update the search

line 113: please add the related reference

Round 2

Reviewer 1 Report

In order to standardize and validate terms related to medication adherence and eliminate ambiguity in adherence search, the authors searched the literature, translated the terms and definitions, and translated and adapted the original ABC classification of medication adherence into Italian through three rounds of Delphi surveys. The article has clear thinking and rigorous structure. If the author can improve the following questions, the article structure will be more complete:

1.What are the main differences in Italy's understanding of terms related to compliance?

2.What is the relationship between patient medication adherence and adherence terminology?

3. The importance of standardized terminology for improving medication adherence is less discussed, and this part can be appropriately added.

      4. Is there a statistical difference in the composition of the number of experts in different professional fields in the groups? If there is a difference, does it affect the result? Is the distribution of the number of experts in different professional fields who accept or reject the definition of each term in each round Delphi survey?
